# Audio-Visual Speech and Gesture Recognition by Sensors of Mobile Devices

**DOI:** 10.3390/s23042284

**Published:** 2023-02-17

**Authors:** Dmitry Ryumin, Denis Ivanko, Elena Ryumina

**Affiliations:** St. Petersburg Federal Research Center of the Russian Academy of Sciences (SPC RAS), 199178 St. Petersburg, Russia

**Keywords:** audio-visual speech recognition, model-level fusion, lip-reading, gesture recognition, spatio-temporal features, dimensionality reduction technique, computer vision

## Abstract

Audio-visual speech recognition (AVSR) is one of the most promising solutions for reliable speech recognition, particularly when audio is corrupted by noise. Additional visual information can be used for both automatic lip-reading and gesture recognition. Hand gestures are a form of non-verbal communication and can be used as a very important part of modern human–computer interaction systems. Currently, audio and video modalities are easily accessible by sensors of mobile devices. However, there is no out-of-the-box solution for automatic audio-visual speech and gesture recognition. This study introduces two deep neural network-based model architectures: one for AVSR and one for gesture recognition. The main novelty regarding audio-visual speech recognition lies in fine-tuning strategies for both visual and acoustic features and in the proposed end-to-end model, which considers three modality fusion approaches: prediction-level, feature-level, and model-level. The main novelty in gesture recognition lies in a unique set of spatio-temporal features, including those that consider lip articulation information. As there are no available datasets for the combined task, we evaluated our methods on two different large-scale corpora—LRW and AUTSL—and outperformed existing methods on both audio-visual speech recognition and gesture recognition tasks. We achieved AVSR accuracy for the LRW dataset equal to 98.76% and gesture recognition rate for the AUTSL dataset equal to 98.56%. The results obtained demonstrate not only the high performance of the proposed methodology, but also the fundamental possibility of recognizing audio-visual speech and gestures by sensors of mobile devices.

## 1. Introduction

Audio-visual speech recognition (AVSR) is a key component of modern human–computer interaction (HCI) systems, especially in acoustically noisy conditions that often occur in mobile devices applications. The general idea is to recognize speakers’ commands in a video based on both audio and video signals. The introduction of visual information can help to localize speakers and recognize speech commands better. Along with this, there is a possibility to use visual information for gesture recognition. Combined audio-visual speech and gesture recognition will lead to significant improvements of friendliness and effectiveness of HCI systems [1].

Automatic speech recognition (ASR) is the most natural, convenient, and user-friendly way of communicating for humans. However, performance of modern ASR systems often significantly degrades in real-world noisy conditions due to mismatch between training and the real environment. Despite many technologies that have been developed in order to achieve noise robustness, most of them fail to do so in real environments with various types of noise [2]. Alternatively, visual information is not distorted by acoustic noise, and automatic lip-reading plays an important role in acoustically difficult conditions.

Usually, when people are listening to speech in an acoustically noisy environment, they perform lip-reading subconsciously for more additional information, which is of great benefit for human speech perception [3,4]. Even in quiet office conditions, seeing the lips of the speaker significantly improves perception, as demonstrated by the famous McGurck effect [5]. Automatic lip-reading generally provides consistent recognition accuracies regardless of signal-to-noise-ratio (SNR), whereas ASR systems usually perform worse with lower SNR [6]. However, it is obvious that acoustically based speech recognition commonly achieves higher recognition accuracy than lip-reading due to audio information providing more sufficient cues to classify phonemes than visual mouth movements. AVSR tries to combine the benefits of both modalities and reduce the shortcomings of each.

Automatic AVSR systems have been developed for many years. However, modern AVSR systems, whether hybrid or end-to-end (E2E), still have a lot of room for improvement in real-life applications.

Along with this, it is well-known that hearing-impaired people are limited in their ability to communicate with hearing people through normal speech. According to the official statistic of the World Health Organization for 2021 (http://www.who.int/mediacentre/factsheets/fs300/en/ accessed on 6 February 2023), there were about 466 million people in the world (more than 5% of the total population of the globe, of which 34 million are children) who suffer from complete deafness or have hearing problems. In addition, one in three people over the age of 65 experience hearing loss, and it is estimated that more than 2 billion people will be deaf or hard of hearing by 2050. Therefore, intelligent technologies (systems) of effective automatic machine sign language recognition (SLR) are needed to organize a natural HCI [7].

One of the main criteria for the successful organization of HCI [8] is the naturalness of communication [9,10]. Ideally, HCI in terms of modality should not differ from interpersonal communication. Therefore, the main feature of modern intelligent systems is the use of methods of communication common between people. Non-verbal interaction (in particular, body language, gestures, facial expressions, and articulation) is an integral part of natural communication [11]. Therefore, for example, using gestures, we can interact with an intelligent information system at some distance and in conditions of strong background noise, when the sounding speech is ineffective [12,13,14]. However, it should be noted that there are still no full-fledged automatic systems for machine SLR. This is due a number of factors (visual noise, occlusions, changes in illumination), insufficient description of the grammar and semantics of sign languages (SLs), as well as a number of other speaker-related features.

The gender and age of a single speaker can affect the size of the gestures, the distance of the hands from the body, the distance between the active and passive hand, and the speed of demonstration of various lexical gestural units or clauses. The influence of gender and age aspects on non-verbal behavior are widely described in work devoted to gender linguistics [15], nonverbal semiotics [16], and psychology [17,18]; however, they are practically not taken into account in the context of machine SLR and computer vision (CV) methods. In addition, deaf people are often known to accompany their gestures with almost silent lip articulation [19]. All this allows for concluding that the task of machine SL recognition is a complex interdisciplinary study and requires fundamentally new scientific and technical results that will allow the most effective recognition of individual gestures, as well as elements of SL.

Thus, we consider two actual problems of computer vision: AVSR and gesture recognition. We offer state-of-the-art deep neural network-based methodology for audio-visual information processing. We train both audio and visual models independently for the AVSR task and perform their fusion at the model-level. This allows us to create an E2E AVSR system, which, like a human brain, simultaneously analyzes two sources of information. We then used a model trained on the visual speech recognition (SR) task to extract features for representing lips in the gesture recognition task. This allows us to combine the two tasks and improve the quality of human–machine interaction using mobile device sensors.

In this article, we present state-of-the-art results on audio-visual speech and gesture recognition. We propose a deep neural network-based model architecture for each task. We benchmark our methodology on two well-known datasets: LRW [20] for audio-visual speech recognition and AUTSL [21] for gesture recognition. We outperformed existing methods on both tasks. The accuracy of AVSR is achieved by fine-tuning the parameters of both visual and acoustic features and the proposed E2E model. The accuracy of gesture recognition is achieved through the use of a unique set of spatio-temporal features, including those that take into account lip articulation information. Our research integrates two complex tasks in computer vision and machine learning: lip-reading and gesture recognition. A thorough review of prior work reveals that this is the first time lip articulation has been used in the problem of gesture recognition.

We emphasize that the use of visual information can significantly improve speech and gesture recognition. To the best of our knowledge, currently there are no such systems that are able to perform both tasks. The results obtained demonstrate not only the high performance of the proposed methodology, but also the fundamental possibility of recognizing audio-visual speech and gestures by sensors of mobile devices.

The remainder of this article is organized as follows: Section 2 summarizes related work on both AVSR and gesture recognition tasks. In Section 3, we describe the datasets used for training, validating, and testing. In Section 4, we propose AVSR and gesture recognition methods and models. Our proposed methods are evaluated and compared in Section 5. Finally, some concluding remarks are presented in Section 6.

## 2. Related Work

Many methods have been proposed for both audio-visual (AV) speech and gesture recognition. It is the task of recognizing both phrases and gestures based on audio and visual information. However, in existing scientific research, these two problems were usually treated separately, so further AVSR and SL will be analyzed in different subsections.

### 2.1. Audio-Visual Speech Recognition

Traditionally, AVSR systems consist of two processing stages: feature extraction from audio and visual information followed by modality fusion and recognition [22,23]. For traditional methods, features are usually extracted around the mouth region-of-interest (ROI) and from the audio waveform and then concatenated [24,25,26]. In traditional methods of AVSR, a transform (e.g., principal component analysis (PCA) [27], linear discriminant analysis (LDA) [28], or t-distributed stochastic neighbor embedding (t-SNE) [29]) is usually applied to the detected ROI for video and concatenated mel-frequency cepstral coefficients (MFCCs) for audio, followed by a deep autoencoder to extract bottleneck features [30,31,32]. Then, extracted features are fed to a classifier such as support vector machine (SVM) [33], hidden markov model (HMM) [34], coupled HMM [35], etc.

In recent years, with the development of deep learning technology, many deep learning methods have been presented and have replaced the feature extraction step with deep bottleneck architectures. The first convolutional neural network (CNN) image classifier to discriminate visemes was trained in ref. [36,37]. In [38], the deep bottleneck features were used for word recognition in order to take full advantage of deep convolutional layers and explore highly abstract features. Similarly, it was applied to every frame of the video in [30]. The authors in Ref. [39] proposed using 3D convolutional filters to process spatio-temporal information of the lips. Then, researchers in Ref. [40] applied an attention mechanism to the mouth ROI and MFCCs.

Finally, E2E architectures have been presented recently for ASR and have attracted a great amount of attention. The main advantage of the modern E2E method is the ability to process both features extraction and classification stages in a single neural network (NN). These methods can be divided into two groups. In the first group, dense layers are used to extract features, and long-short term memory layers (LSTMs) are responsible for modeling the temporal dynamics [41,42]. In the second group, CNNs are used for feature extraction followed by LSTMs or gated recurrent unit layers (GRUs) [43].

Recently, E2E methods have been successfully used for many SR [44,45], emotion recognition [46], and CV tasks [47]. However, research on E2E AVSR or gesture recognition has been very limited. We could note works [48,49] where attention mechanism was applied to both the lip ROIs (video) and MFCCs (audio) and the model was trained E2E. Then, fully connected layers followed by LSTMs are used to extract features from images and spectrograms and perform classification.

The first E2E model that performed AV word recognition from raw mouth ROIs and waveforms on a large in-the-wild dataset was Ref. [50]. The authors proposed a two-stream model for features extraction. Each stream consisted of ResNet [51], which extracts features from the raw input, followed by a 2-layer bidirectional GRU (BiGRU), which models temporal dynamics in each modality stream. In order to build an E2E network, researchers in Ref. [41] used LSTMs to extract features from the raw data. Usually, existing methods take the mouth region as a whole, however, researchers in Ref. [1] proposed to use separate lip parts. Researchers in Ref. [52] compared and analyzed AVSR models by applying either cross-entropy loss or connectionist temporal classification (CTC) loss to a transformer-based AVSR model.

However, AV modality fusion mechanisms should be still developed to achieve successful recognition performance in both acoustically clean and noisy conditions. In the work [48], modality attention computes scores for modality space in order to train attention with balanced modalities. Modality attention is usually applied when audio and visual modalities have the same time length. However, audio and visual features are usually generated at different time steps and have to be resampled [53].

A transformer model was initially proposed in machine translation [54], and, since then, there have been many studies to introduce the transformer model not only to ASR but to AVSR. The transformer model calculates the global context over the entire input data, which might result in improved performance and faster and more stable training [55,56]. That is the main difference with LSTM- and Bi-LSTM models that compress all input data into a fixed-size vector. In Ref. [57], the transformer model was also combined with the LSTM-based model. In a typical AVSR transformer model, there are two encoders for audio and video and one common decoder. Recently, an efficient fusion method of audio and visual in a transformer-based AVSR model was also proposed [58].

In order to develop noise-robust ASR systems, high-quality training and testing datasets are crucial. Regarding available AV speech datasets, the are many collected for different purposes and with different means. The researchers in works [59,60] provide comprehensive analysis on existing AV speech datasets. Combining video and audio information can improve SR accuracy for low signal-to-noise ratio conditions [61]. It has been demonstrated that, for humans, the presence of the visual information is roughly equal to a 12 dB gain in acoustic signal-to-noise ratio [62].

Another modern trend that appeared recently is the web-based datasets: datasets collected from open sources such as YouTube or TV shows [59]. The most well-known of them are: LRW dataset [20], LRS2-BBC, LRS3-TED datasets [63], VGG-SOUND [64], Modality dataset [65], and vehicle AVSR [66]. A survey [67] regarding this topic provides essential knowledge of the current state-of-the-art situation.

The combination of state-of-the-art deep learning methods and large-scale audio-visual datasets has been highly successful, achieving significant recognition accuracy results and even surpassing human performance. However, there is still a long way to go for practical AVSR applications to meet the performance requirements of real-life scenarios.

### 2.2. Gesture Recognition

In the last decade, scientists have been actively conducting scientific and technical research (especially in the field of CV) and developing new technologies for automatic recognition of the SL of deaf people: Keskin C. [68,69,70], Akarun L. [71,72,73], Koller O. [74,75,76,77,78,79,80], etc. [81,82,83,84,85,86,87,88,89,90,91,92,93,94]. The originality of the selected scientific studies is highlighted below.

In Ref. [68], the authors presented a method for hand pose estimation and hand shape classification using a multi-layered randomized decision forest algorithm. In a follow-up study [69], the authors proposed a real-time method for capturing hand posture using depth sensors and a 3D hand model with 21 parts and a random decision forest for pixel classification and joint location estimation. Another relevant work [70] proposed a generative model and depth data-based method for hand tracking, using an articulated signed distance function to model hand geometry for fast optimization and high frame rates. This system was capable of tracking two hands interacting with each other or objects.

The work presented in Ref. [71] focused on the development of a real-time CV system for aiding hearing-impaired patients in a hospital setting. The system engages users through a series of questions to determine the purpose of their visit and elicits responses through SL. In Ref. [72], the authors propose the use of temporal accumulative features for recognizing isolated SL gestures. This method incorporates SL-specific elements to capture the linguistic characteristics of SL videos, resulting in an efficient and quick SL recognition system. In Ref. [73], the authors introduce a method for translating SL into written text using NNs and a learning-based method for tokenization. The authors aim to improve SLR and translation systems by incorporating a tokenization step to better capture the linguistic structure of SL.

In Ref. [74], the authors presented a method for translating SL into written text using NNs. The study aimed to capture the linguistic structure of SL through a NN-based method, with the findings offering insights into the potential of NNs to improve SLR and translation systems. Another study [75] explored weakly supervised learning for SLR using a multi-stream CNN-LSTM-HMM model to uncover the sequential parallelism in SL videos. The authors trained the model with weakly labeled data, demonstrating the potential of weakly supervised learning to enhance SLR and translation systems. In Ref. [76], the authors addressed the challenge of multi-articulatory SL translation by proposing a multi-channel transformer architecture. This architecture enables the modeling of inter- and intra-contextual relationships between different signers while preserving channel-specific information. The authors of Ref. [77] proposed an E2E joint architecture based on the transformer network for SLR and translation. The architecture merges the recognition and translation tasks into a single model, significantly improving performance compared to conventional methods that undertake recognition and translation as separate processes. In Ref. [78], the authors examine the challenges of gathering SL datasets for training machine learning models, including privacy, participation, and model performance. The study provides valuable insights into the complexities of collecting high-quality SL data and highlights the importance of considering privacy and ethical concerns in SL research. An interdisciplinary study in Ref. [79] provided a comprehensive overview of SL datasets. The study categorized datasets based on factors such as modality, language, and application, and provided an analysis of each dataset and its suitability for various SLR tasks. The authors also discussed the limitations of current datasets and suggested future directions for improvement, making it an important resource for researchers and practitioners in the field of SLR. In addition, the authors of Ref. [80] presented Microsoft’s submission to the workshop on statistical machine translation shared task on SL translation, which utilized a clean text and full-body transformer model. The aim of the research was to improve the translation of SL into written text through this methodology.

In Ref. [81], the authors provided a comprehensive review of hand gesture recognition techniques, including CV-based methods, machine learning algorithms, and wearable device-based methods. In Ref. [82], a method that combines 3DCNN and convolutional LSTM for multimodal gesture recognition was presented, showing the effectiveness of such a combination. The authors in Ref. [83] proposed a method that improves dynamic hand gesture recognition using 3DCNNs by embedding knowledge from multiple modalities into individual networks. In Ref. [84], MultiD-CNN, a multi-dimensional feature learning method for RGB-D gesture recognition using deep CNNs, was proposed. The authors in Ref. [85] presented a method for gesture recognition using multi-rate and multi-modal temporal enhanced networks, which employ a search algorithm to determine the optimal combination of network architecture, temporal resolution, and modality information. The study in Ref. [86] reviewed gesture recognition in robotic surgery, whereas Ref. [87] presented a real-time hand gesture recognition system using YOLOv3, and Ref. [88] presented a multi-sensor hand gesture recognition system for teleoperated surgical robots. In Ref. [89], the authors showed the feasibility of hand gesture recognition using electromagnetic waves and machine learning. The authors in Ref. [90] explored ensemble methods for isolated SLR, and in Ref. [91], a sign pose-based transformer method for word-level SLR was proposed. In Ref. [92], a SLR method that utilizes a palm definition model and multiple classification was presented, and in Ref. [93], an ensemble method using multiple deep CNNs was presented for SLR. In Ref. [94], a method for few-shot SLR using online dictionaries was proposed.

Scientists from Carnegie Mellon University should also be noted, as they were among the first to develop an open-source solution (OpenPose (https://cmu-perceptual-computing-lab.github.io/openpose/web/html/doc/index.html accessed on 6 February 2023)) to determine multiple skeletal and facial landmarks (human skeletal model) in individual images in real time. A detailed description of the OpenPose library is presented in [95,96,97]. At the same time, Google is actively developing a cross-platform open source environment MediaPipe (https://google.github.io/mediapipe/ accessed on 6 February 2023), which includes new methods based on deep learning to determine three-dimensional (3D) landmarks of the face [98,99], hands [100], and body [101] of a person. In turn, the scientific and technical group from Meta AI Research (https://ai.facebook.com accessed on 6 February 2023) presented the FrankMocap [102,103] library, focused on 2D localization of the area (including the areas of the hands) with its further 3D visualization in real time.

To date, the scientific community and large technical corporations have collected and annotated many visual and multimodal datasets for solving problems of both localization of human facial and skeletal landmarks and recognition of SL (for example: LSA64 (http://facundoq.github.io/datasets/lsa64/ accessed on 6 February 2023)) [104], MS-ASL (https://www.microsoft.com/en-us/research/project/ms-asl/ accessed on 6 February 2023)) [105], CSL (http://home.ustc.edu.cn/~pjh/openresources/cslr-dataset-2015/index.html accessed on 6 February 2023)) [106], TheRusLan [107], AUTSL [21], WLASL (https://dxli94.github.io/WLASL/ accessed on 6 February 2023)) [108], and WLASL-LEX [109]). Portions of them are publicly available and free for research experiments.

Thus, all studies are aimed at solving the problems of effective complex intellectual analysis of human body movements for automatic recognition of SL. However, it is worth noting that it is still quite difficult to completely abstract from the digital scene (video information) and analyze only the dynamically changing state (behavior) of a person (including SL). There are currently no fully automatic NN models and methods for machine SLR systems. To create such full-fledged NN models, it is necessary to perform a deep intellectual analysis and improve methods for extracting not only spatial, but also temporal features from a localized area with a person.

## 3. Research Datasets

For the purpose of this study, we use two large-scale publicly available datasets: the Lip Reading in the Wild (LRW) (https://www.robots.ox.ac.uk/~vgg/data/lip_reading/lrw1.html accessed on 6 February 2023) [20] for AVSR and the Ankara University Turkish Sign Language dataset (AUTSL) [21] for gesture recognition. Both datasets are very challenging, as there are large variations in head pose, illumination, acoustic conditions, etc.

### 3.1. Audio-Visual Speech Recognition

The LRW [20] is a large, publicly available dataset. The dataset consists of short segments (1.16 s) from BBC programs, mainly news and talk shows. It is a very challenging set because it contains more than 1000 speakers. The number of recognition classes is 500. This number is much higher than existing audio-visual datasets, which typically contain 10 to 50 classes. The LRW main characteristics are presented in the Table 1.

Another characteristic of the dataset is the presence of several words that are visually similar. For example, there are words that are present in their singular and plural forms or simply different forms of the same word, e.g., America and American. It is worth noting that the words are not isolated: they are taken in-the-wild conditions, so some co-articulation of the lips from preceding and subsequent words is present.

### 3.2. Gesture Recognition

All modern multimodal datasets differ in the number of movements (gestures), video capture hardware, background environment, and, most importantly, the tasks for which they were created. Most of the datasets are designed for the tasks of recognizing individual gestures and movements. In the current study, we use the AUTSL [21] large-scale multimodal Turkish sign language dataset. The main differences between AUTSL [21] and many other datasets are as follows:Multimodality (video data in RGB format with depth map);All gestures are rendered dynamically;Quite a large number of signers (43 people);Quite a large number of gestures (226 Turkish SL gestures);Various background settings.

Figure 1 shows examples of images of the faces of all signers from the AUTSL [21] dataset.

As can be seen from Figure 1, the distribution of signers by gender is 10 male to 33 female. In addition, from the description of the competition held in 2021 as part of the CVPR conference “Looking at People Large Scale Signer Independent Isolated SLR CVPR Challenge” (https://chalearnlap.cvc.uab.cat/dataset/40/description/ accessed on 6 February 2023), it is known that the age of signers varies from 19 to 50 years, and the average age of signers is 31 years. In addition, within the framework of this study, the total number of signers who accompany gestures with lip articulation, as well as the number of gesture repetitions per signer and other statistical characteristics (see Table 2), was calculated.

As we can see from Table 2 in the Train and Val samples, most of the signers accompany the gestures with lip articulation. In turn, the Test set is balanced in relation to articulated and non-articulated signers. Therefore, it can be assumed that the gender and age characteristics of the signers, together with the signs of their lip articulations, can affect the accuracy of machine SL translation.

## 4. Methodology

In this section, we describe proposed methods to AVSR and gesture recognition. We illustrate in detail the proposed pipeline and models architecture.

### 4.1. Audio-Visual Speech Recognition

Figure 2 demonstrates the proposed audio-visual method for SR. The method uses two open source libraries: MediaPipe Face Mesh [110] for video pre-processing and Librosa [111] for audio pre-processing.

Initially, images of the lip region are extracted using the MediaPipe Face Mesh [110] algorithm. Due to the influence of the articulation, the shape of the lips, the proportions of the face, etc., all images have to be normalized to a size of 44 × 44 × 3 by padding the missing pixels with average values. Because each video has 29 frames per second, the sequence length is 29 images. Using Librosa [111], a log-Mel spectrogram image with 64 Mel filter-banks is extracted from the audio signal, with a short-time Fourier transform window size of 2048 and a step of 64. The resulting image has a dimension of 64 × 305 × 3. Min–max normalization is applied to images.

The images are fed to the audio-visual model. It consists of two separate parts for processing audio and video signals. Both of them are based on the ResNet-18 [51] model architecture. The visual model produces a feature vector with the size of 1024; the audio model has an output feature vector with the size of 512. Then, the feature vectors are concatenated into one vector and fed to the final fully connected neural network (FCNN) to make a prediction. Both models were trained with the same parameters: learning rate, schedule, optimizer, and batch size. Simultaneous training of two models accounts for the benefits of both modalities. Therefore, our audio-visual model works like the human brain, analyzing both acoustic and visual information simultaneously. This strategy is known as model-level fusion [112].

The choice of models’ architectures in the proposed method was based on a series of experiments, which are described in detail in the following sections.

#### 4.1.1. Visual Speech Recognition

In order to choose a visual SR model, we carefully studied the state-of-the-art methods proposed for the LRW dataset [20]. Most of the existing methods are based on the ResNet-18 [51] model. In this research, we implement three different models based on ResNet-18 [51] architecture: 2DCNN+BiLSTM, 3DCNN, and 3DCNN+BiLSTM. The architectures of the three implemented models are shown in Figure 3.

All three compared models analyze the sequence of frames and their spatio-temporal dependencies. However, they have fundamental differences. The 2DCNN+BiLSTM model consists of static (2DCNN) and spatio-temporal (BiLSTM) models. 2DCNN can process B × W × H × C input data, where B is the batch size, W is the image width, H is the image height, and C is the number of image channels. Whereas BiLSTM works with feature dimensions B × T × F, where T is the length of the feature sequence. At the same time, we feed the input data with dimensions B × T × W × H × C to the input of 2DCNN+BiLSTM. In order to ensure the processing of sequences, the TimeDistributed layer is used, which allows combining outputs from the 2DCNN layer for each sequence in a batch, i.e., 2DCNN with the same weights is applied as many times as the dimension of one batch. The number of parameters of such a model is slightly more than 22 million. The 3DCNN model does not require additional spatio-temporal models, as it is itself capable of processing image sequences and their depth. At the same time, such a model has more parameters—more than 34 million. Such a number of parameters is due to the fact that 3DCNN works with the global temporal information and local spatio information of the input data [113].

Finally, if we do not reduce the depth of the input dimension (by setting the depth stride to 1, for example (2,2,2)/2 set (1,2,2), see Figure 3), then for sequential processing, we use the BiLSTM model, and this eliminates the need to use the TimeDistributed layer. The 3DCNN+BiLSTM model has about 44 million parameters. Thus, the 2DCNN+BiLSTM model studies spatio-temporal information only at the BiLSTM level, the 3DCNN model—at the convolution level, and 3DCNN+BiLSTM—at both levels.

#### 4.1.2. Audio Speech Recognition

Log-Mel spectrograms are widely used in deep learning for various CV tasks such as speech escalation detection [114], audio classification [115], and ASR [116]. In the current work, we also use log-Mel spectrograms, and as deep learning models we implement three 2DCNN models: ResNet [51], VGG [117], and PANN [118]. The selected models have been repeatedly used in CV tasks for audio modality processing [114,116]. Model architectures are shown in Figure 4. Each of the three models consists of a sequence of convolutional blocks (see Figure 4). The architectures differ in the number of repetitions of convolutional blocks, filter sizes, and, consequently, the number of parameters. The ResNet model [51] has over 11 million parameters, PANN [118] has about 5 million, and VGG [117] has about 15 million. We compare these three models and choose the one that shows better performance in the ASR task.

#### 4.1.3. Audio-Visual Fusion

Previously, we described models for uni-modal SR based on video or audio speech processing. However, the use of one modality in real conditions has a number of limitations: malfunction of cameras or microphones, data noise, lighting instability, face occlusion, etc. At the same time, the combination of modalities allows use to compensate for their shortcomings. In this study, we implemented three fusion strategies and compared their performance. Figure 5 illustrates the analyzed modality fusion strategies.

The prediction-level fusion is the simplest strategy to implement. We get predictions from the trained models for the Val and Test set of the LRW dataset [20]. From each modality, we get a vector of predictions equal to 500 (there are 500 classes in the LRW dataset [20]). We use the weighted prediction fusion method, which has shown its effectiveness in other CV problems [119,120,121]. To obtain weighted predictions, we use the Dirichlet distribution to form a tensor with dimensions of 1000 × 500 × 2, where 1000 is the number of randomly generated 500 × 2 weight matrices, 500 is the number classes, and 2 is the number of models. First, the best matrix on the Val set of the LRW dataset is selected. Then this matrix is applied to the Test set to form the final vector and determine the class with the highest probability.

The feature-level fusion, unlike the previous strategy, requires the use of additional trained models to study feature relationships both within one modality and within two modalities. For feature-level fusion strategies, both traditional models [122] and NN models [53] are used. We use a conventional FCNN that takes a combined feature vector as input and produces a final prediction vector for 500 classes.

In model-level fusion, one common model is trained. We combine the two best audio and video models, initialize their weights, and jointly fine-tune them. Such a strategy, as noted earlier, works like the human brain, which is able to simultaneously analyze visual and acoustic information.

### 4.2. Gesture Recognition

Hand gestures refer to a non-verbal way of communicating and allow for conveying thoughts, feelings, and emotions of a person. Each individual hand gesture has its own structure [123] formed from its individual elements [124]. Each gesture also has a constant characteristic in the form of the shape of the hand, the location of the gesture in space, and the nature of [125] execution. The hand configuration describes a specific palm position and finger direction [126,127]. The location of a gesture in space is necessary to determine the semantic meaning of a gesture, as the localization of all gestures is always strictly constant. The nature of the gesture performance depends on its static or dynamic reproduction by the signer. A static gesture consists of a stable shape of the hand in time and space, whereas the configuration of a dynamic gesture is variable, both in time and space. It is also worth considering the fact that during the demonstration of a gesture by the signer, the general understanding is made up of many movements of the hand(s). For example, the usual handshake varies not only from person to person, but also depends on time and space.

Thus, in a broader sense, for recognition of a static gesture, it is necessary to focus on determining the shape of the hand, whereas for a dynamic gesture, it is worth focusing on the movement of the hand. Dynamic gestures consist of the following steps:Preparing a gesture;Functional component of the gesture (its core);Retraction [128].

Gesture preparation may consist of initial hand direction to the start of the gesture, neutral hand movement, or residual movement from a previous gesture. The functional core of the gesture includes context-independent hand movement in relation to other gestures. Retraction should be understood as the movement of the hand to prepare for the next gesture. However, it is worth noting that there is a problem with each signer showing gestures at different speeds. That is why almost all modern gesture recognition methods are reduced to processing a video sequence that provides information about the movements of any part of the human body, for example, a hand or both hands in time and space [129,130,131,132,133,134]. Additionally, the presence of complex background situations on video frames that dynamically change leads to rather serious recognition problems due to insufficient use of the spatial features: hand gestures are relatively small in size compared to the entire background environment. In addition, tasks for recognizing gestures of any SL are also characterized by other important parameters:Size of recognition dictionary;Variation of signers (gender and age) and gestures;Characteristics of the visual information transmission channel.

The lexical components of SL (complete hand gestures) are formed from several components:Hand configuration (shape of hand or hands);Place of performance (hands in space during the gesture);The nature of the movement;Facial expressions;Lip articulation.

That is why it is reasonable to build the process of recognition of gestures taking into account their spatio-temporal component. In this regard, we propose our method for recognizing gestures, which is based on spatio-temporal features (STF). The proposed method is shown in Figure 6.

According to Figure 6, the input video file goes to the landmark detection model. We use MediaPipe Holistic (https://google.github.io/mediapipe/solutions/holistic.html accessed on 6 February 2023), which combines separate NN models to determine 2D landmarks of the face [98,99], hands [100], and human [101] bodies. Based on the obtained landmarks (see Figure 7b), graphic features are calculated, including:2D distances from face to hands are calculated as:
(1)$$dist=\sqrt{{\left(x_{f}-x_{h}\right)}^{2}+{\left(y_{f}-y_{h}\right)}^{2}},$$
where dist is the 2D distance between the face and hand (right or left); $x_{f}$ and $x_{h}$ are the *x* coordinates of the face and hand, respectively; and $y_{f}$ and $y_{h}$ are the *y* coordinates of the face and hand, respectively. We take into account the upper right point of the face region (see Figure 7c, orange box) and left point of the hand region (see Figure 7c, blue box) to calculate the distance between the face and the left hand. For the right hand, the distances are calculated from the upper left point of the face and right point of the hand (see Figure 7c, green frame);Areas of face and hands intersection are calculated as:
(2)$$\tilde{x}=\left\{\begin{array}{l} 0,\;\;if\;x_{end}-x_{start}\leq 0,\\ x_{end}-x_{start},\;\;else, \end{array}\right.$$
(3)$$\tilde{y}=\left\{\begin{array}{l} 0,\;\;if\;y_{end}-y_{start}\leq 0,\\ y_{end}-y_{start},\;\;else, \end{array}\right.$$
(4)$$Area_{intersection}=\tilde{x}\cdot \tilde{y},$$
where $\tilde{x}$ and $\tilde{y}$ are intersection width and height; $x_{end}$ is min value of two max *x*-coordinates of two bounding boxes (face and hand); $x_{start}$ is max value of two min *x*-coordinates; $y_{end}$ and $y_{start}$ are min and max values of two max and min *y*-coordinates, respectively; and $Area_{intersection}$ is area intersection. If there is no intersection, the area will be zero;Zones of hands location, which are illustrated in Figure 7d. The presented zones (five zones) for showing gestures make it possible to describe all available gestures in the *Y*-plane. The area with the hand belongs to one of the five gesture zones if the area of their intersection is greater than 50%. In rare cases, when an area with a hand intersects simultaneously by 50% with two of the five zones, then the zone is selected by its smallest initial coordinate ($y_{min}$) relative to the *Y*-plane.

All three graphic features are calculated for each hand of each frame. Thus, a total of six graphic features (two hands) are extracted per frame. These signs characterize changes in the position of hands in 2D space relative to the face and the zone of their demonstration.

Also, based on previously obtained landmarks, a search for ROI is performed, including: the face regions for each frame, lips, and hands. ROIs are shown in Figure 7c. The face region is fed to pre-trained models (https://github.com/serengil/deepface accessed on 6 February 2023) from the Deepface open source software platform [135,136] for machine classification of the signer’s gender and age. Previously, we used these models in a similar problem of gesture recognition [137]. In Ref. [107], an increase in accuracy was obtained by considering gender and age of the signer [138] (91.14% vs. 88.92%, gain 2.22%). Gender is represented by the numeric value of the class (0—”male”, 1—”female”). Age is presented in the range from 1 to 100 years.

We extract NN representations from the lip regions using the model developed (2DCNN+BiLSTM) in the current article for automatic lip-reading. Even though we trained our model on the English lip recognition task, the model can be used to recognize the speech of other languages. This strategy is called transfer learning and has proven effective in other CV problems [120]. Finally, we extract NN hand representations using the E2Ev2 [137] model.

Both NN models analyze frame sequences and have two layers of LSTM (in the E2Ev2 model) or BiLSTM (in the 2DCNN+BiLSTM model). To obtain features for all frame sequences, we extract them from the first layers, because the second layers produce one feature vector per sequence. In this regard, for one image of the lips, we get a vector of features with a dimension of 1024 (corresponding to the output of the first BiLSTM layer of the 2DCNN+BiLSTM model for one frame), and for the image of each hand—512 (corresponding to the output of the first LSTM layer of the E2Ev2 model for one frame). This number of features greatly exceeds the number of other proposed features in our gesture recognition method, so we use and compare some dimensionality reduction techniques (DRT), namely: PCA [27], LDA [28], and t-SNE [29]. The main idea of PCA is to maximize the variability (dispersion) of the data by performing linear combinations on features. The idea behind LDA is to maximize the dispersion between different classes and minimize the dispersion within a class. The t-SNE technique does not rely on the dispersion of features; it tries to find their two-dimensional representation, which will preserve the distance between feature points as much as possible. PCA and t-SNE are unsupervised dimensionality reduction techniques. A comparison of the techniques used to reduce feature dimensionality is presented in the experimental results.

Thus we form seven types of STF. The STF are then combined into a single vector, normalized by Z-normalization, and fed into a gesture recognition model. The architecture of the gesture recognition model is shown in Figure 8. The gesture recognition model consists of two BiLSTM networks of 64 and 32 units with an attention layer between them. Attention was proposed in [139] and tested on other CV problems [140]. The FCNN completes the gesture recognition model and predicts 226 gestures.

## 5. Evaluation Experiments

In this section, we present the results of evaluation experiments on the (1) selection of optimal models, (2) input image parameters, and (3) augmentation techniques for SR based on video and audio data processing. We evaluate the experiments to optimize the gesture recognition model.

### 5.1. Audio-Visual Speech Recognition

Here we present the results of SR for audio and video modalities and the fusion of both modalities.

#### 5.1.1. Visual Speech Recognition

To build a reliable model for visual SR, we conduct a series of experiments that can be divided into the following groups:The selection of model architecture;The selection of optimal input image resolution;The selection of optimal data augmentation methods [128].

The first group of experiments is presented in Table 3. We compare three 3DCNN, 2DCNN+BiLSTM, and 3DCNN+BiLSTM models (see Figure 3), which we train considering:Two learning rate schedulers (constant learning rate, cosine annealing learning rate [141]). The learning rate on cosine annealing is calculated as:
(5)$$lr=\frac{lr_{start}}{2}\cdot \left(cos\left(\frac{mod\left(epoch_{curr}-1,\left[\frac{epochs}{cycles}\right]\right)}{\left[\frac{epochs}{cycles}\right]}\right)+1\right),$$
where $lr_{start}$ is the initial learning rate, cos() is the cosine of the value, mod() is the remainder of division, $epoch_{curr}$ is the current epoch, epoch is the number of epochs, and cycles is the number of learning rate restart cycles. Learning rate restart cycles are set to one hundred epochs;Two optimizers (Adam, SGD). The maximum accuracy of SR for the Adam optimizer is achieved at a learning rate 10 times less than with the SGD optimizer.

The following basic parameters were set for all models: (1) image resolution—88 × 88 × 3; (2) image pixels are padded with average values if the image resolution is less than the set one; (3) batch size—4. For these experiments, the number of training epochs is set to 100; however, training stops if the recognition accuracy on Val set does not increase within 6 epochs. We train all models from scratch, because the LRW dataset [20] has about 800–1000 instances of training data for each class, so there is no need to apply transfer learning.

The experimental results presented in Table 3 demonstrate that the accuracy obtained by the 2DCNN+BiLSTM and 3DCNN+BiLSTM models is almost 2% higher than the accuracy of the 3DCNN model. This is likely achieved through the use of the BiLSTM model. The 3DCNN+BiLSTM model is slightly inferior to the 2DCNN+BiLSTM model, while the architecture of the second model has two times fewer parameters (22 million versus 44 million). Thus, the 2DCNN+BiLSTM model is the most efficient. Additionally, according to Table 3, we can conclude that by using the cosine annealing learning rate scheduler with the Adam optimizer we gain at least 2% accuracy increase for all models.

The following experiments on selecting the optimal resolution of the input image are carried out using the best 2DCNN+BiLSTM model, the cosine annealing learning rate scheduler, and the Adam optimizer with an initial learning rate of 0.0001. The results of the experiments are presented in Table 4.

The results of the experiments presented in Table 4 demonstrate that the accuracy of visual SR is maximum with image resolution of 44 × 44 × 3 on the LRW dataset. This result is due to the fact that most of the lip images do not exceed the size of more than 50 pixels (both in the image width and in its height). We also compared two image normalization techniques: padding image pixels of average values, or resizing an image to a given size. The results of the experiments showed that when the image is resized, the accuracy of SR drops by 1.4%, probably because the articulation of the lips is distorted with such normalization. We also experimented with the batch size, setting values from 2 to 12. The results of the experiments showed that when batch 2 or 4 was set, we got the same SR accuracy, which was 86.24. At the same time, with an increase in the batch size by 4, the recognition accuracy decreases by approximately 1% each time.

Finally, we analyze how training data augmentation affects the accuracy of video SR. We use training data augmentation techniques such as:MixUp [142] allows mixing two images and their labels with different probabilities. The MixUp is applied to both images and binary vector, and the new image and their label vector are calculated as:
(6)$$\tilde{I}=\lambda \cdot I_{1}+\left(1-\lambda \right)\cdot I_{2},$$
(7)$$\tilde{V}=\lambda \cdot V_{1}+\left(1-\lambda \right)\cdot V_{2},$$
where $\tilde{I}$ and $\tilde{V}$ are the new image and label vector, λ is the coefficient of mixing two images and binary vectors, $I_{1}$ and $I_{2}$ are the first images from the first sequence and the second images from the second sequence, and $V_{1}$ and $V_{2}$ are the binary vector of the first sequence and the binary vector of the second sequence. Two sequences are selected randomly. The λ is set randomly in the range from 0.3 to 0.7 and is applied to all images of sequences. Binary vectors are common to the entire sequence, so the λ is applied only once;Label smoothing [143] softens hot image label vectors. The label smoothing is applied to all binary vectors to which the MixUp augmentation technique has not been applied and is calculated as:
(8)$$\tilde{V}=\left(1-\alpha \right)\cdot V+\frac{\alpha }{K},$$
where $\tilde{V}$ is the new label vector, α is the coefficient responsible for the degree of binary vector smoothing, *V* is the original binary vector, and *K* is the number of classes;Affine transformations are aimed at modifying training images by horizontal and vertical shifts, horizontal flips, shear angle in the counter-clockwise direction, and rotations.

The point of the third technique is to add variation to the training data. The first two techniques are used to make trainable models less confident in their predictions [140], therefore, such models make fewer gross errors, which leads to an increase in the accuracy of SR. The results of experiments on the use of data augmentation techniques are presented in Table 5.

Table 5 shows that using data augmentation techniques can improve accuracy by 1%. At the same time, a greater increase in accuracy is achieved through the use of affine transformations. It is worth noting that earlier we achieved an accuracy of 88.7% [144]; however, unlike the previous work, in the current 2DCNN+BiLSTM model, we do not use the attention module [145], as the use of this module leads to an increase in the number of parameters, which makes it very difficult to be trained and used on mobile devices.

#### 5.1.2. Audio Speech Recognition

Similar to the video modality experiments, we divide the audio experiments into the same three groups. We first compare three ResNet, PANN, and VGG models (see Figure 5), which we train with:Two learning rate schedulers (constant learning rate, cosine annealing learning rate);Two optimizers (Adam, SGD).

For experiments, the following basic parameters were set for all models: (1) the number of Mels—128; (2) the step size of the short-time Fourier transform window—512; (3) the number of image channels is 3; (4) batch size—4. The results of the experiments are presented in Table 6.

Table 6 shows that the ResNet and VGG models handle the task of SR from audio most effectively, where the accuracy obtained using the ResNet model slightly exceeds the accuracy of the VGG model. In addition, unlike the video modality, we can see that the SGD optimizer is in some cases more efficient than the Adam optimizer. Further, the experimental results confirm the efficiency of using the cosine annealing learning rate scheduler; the results of the PANN model especially illustrate this.

The next group of experiments is aimed at identifying the optimal parameters for the log-Mel spectrogram. Experiments are performed using the ResNet model, cosine annealing learning rate scheduler, and SGD optimizer with an initial learning rate of 0.0001. It is worth noting that we are experimenting with two main parameters: (1) the number of Mels; (2) the step size of the short-time Fourier transform window. These options affect the size of the input image for the NN. The results of the experiments are presented in Table 7.

As can be seen from the results of Table 7, when setting the optimal parameters for the log-Mel spectrogram, it is possible to achieve an increase in accuracy by almost 3%. It is best to use a single-channel image of the spectrogram. Additionally, we conducted experiments with the batch size, setting values from 2 to 12. The results of the experiments showed that when batch equals 2, the accuracy is 95.16%, 8—94.51%, 12—93.41%. Therefore, 2 and 4 batches have approximately the same accuracy, whereas with a subsequent increase in the batch size by four, the recognition accuracy decreases by approximately 1% each time. These results are similar to those for video modality.

Finally, we used the number of Mels and a step size of 64 to perform subsequent experiments to augment the training data. For audio modality we used: (1) MixUp [142], (2) SpecAugment [116], and (3) label smoothing [143]. SpecAugment masks the frequency and time scale of the log-Mel spectrogram, which allows simulating microphone dysfunction at a certain time or signal loss at a certain frequency bands due to echo. The results of the experiments are presented in Table 8.

With the help of audio data augmentation techniques (see Table 8), the accuracy of SR is increased by 0.5%, mostly achieved through the use of label smoothing. It should also be noted that augmentation using SpecAugment by frequency (freq mask) does not lead to an increase in the accuracy of SR.

#### 5.1.3. Audio-Visual Fusion

Table 9 presents the accuracy results obtained by fusing both audio and visual modalities. As we can see from the table, the model level fusion allows us to achieve an accuracy of 98.76% on the test set. Feature level fusion is 0.32% inferior to the model level fusion. Prediction level fusion performs worse than the two other strategies. Unlike the other two strategies, prediction level fusion does not have information about the interconnectedness of features obtained from different modalities. It only analyzes the contribution of each modality separately based on the obtained predictions.

Table 9 also demonstrates the accuracy results of modern state-of-the-art methods. It can be observed that the proposed method achieves the highest results in AVSR on the LRW dataset known in the scientific literature to date.

### 5.2. Gesture Recognition

Experiments to optimize the gesture recognition model are focused on the selection of a set of spatio-temporal features. As basic features, we use the [137] features that have proven themselves in our previous study:2D distances from face to hands (two features per frame);Areas of face and hands intersection (two features per frame);Zones of hands location (two features per frame);Age estimate (one feature per frame);Gender estimate (one feature per frame).

To these features, we add hand configuration features extracted using the E2Ev2 [137] neural model. Each hand has its own NN features with the size of 512. Next, the dimension of NN features is reduced using PCA [27], LDA [28], and t-SNE [29]. For PCA [27] and LDA [28], we experimented with component values: 2, 5, 10, 15. Thus, the maximum number of features for hand configurations is 30 (15 for each hand); the minimum is 4. The t-SNE technique [29] allows us to reduce the dimension only to two components. The results of the experiments are presented in Table 10. Here and below, the models are trained on 100 epochs with the Adam optimizer at a rate of 0.00001. Training is interrupted if the recognition rate on the Val set of the AUTSL dataset does not increase within 10 epochs. Recognition rate *r* is used as a performance measure of models for SLR and calculated as:(9)$$r=\frac{1}{N}\sum_{i=1}^{N}f\left(p_{i},t_{i}\right),$$
(10)$$f\left(p_{i},t_{i}\right)=\left\{\begin{array}{ll} 1, & \;\;if\;p_{i}=t_{i},\\ 0, & \;\;else, \end{array}\right.$$
where *N* is the total number of samples, $p_{i}$ is the predicted label for the *i*th sample, and $t_{i}$ is the true label for the *i*th sample.

As we can see from Table 10, the maximum recognition rate of 97.19% is achieved using the LDA dimensionality reduction technique with 10 components. This result is explained by the fact that hand configurations are important features for the task of recognition of gestures, as they contain basic information (flexion of fingers, finger contacts, changes in the number of active fingers), and the more features we analyze, the higher the recognition rate. However, when increasing the number of dimensionality reduction components (up to 15), we did not get an increase in recognition rate. The effectiveness of the LDA technology is explained by the fact that this method reduces the dimension of the feature space based on the labels, i.e., it is a controlled dimensionality reduction technique. The recognition rate without adding the representation of hands is 69.91% (using only basic features). By expanding the set of features, we got an absolute increase in recognition rate equal to 27.28%.

Then, to the already existing set of features (28 features), we add the lip region representation features extracted by the 2DCNN+LSTM model, previously used for lip-reading. We also reduce the dimension of the feature space using PCA, LDA, and t-SNE with the same component values. We report the results for the entire test set and for those speakers who articulate during the gestures. The results of the experiments are presented in Table 11.

Table 11 demonstrates that the 5-component LDA dimensionality reduction technique is sufficient to represent lip regions. The gesture recognition rate for the entire test set was 98.56%, which is 1.34% higher than the recognition rate obtained without using features to represent lip regions. At the same time, the gesture recognition rate for articulating speakers was 99.59%, whereas without taking into account articulation, the gesture recognition rate for the same speakers is 96.99%. Thus, due to the expansion of the set of STF, the gesture recognition rate for articulating speakers increased by 2.57%.

Therefore, our set of STF consists of 33 features:8 basic features;20 features to represent hand configurations;5 features to represent lip regions.

Table 12 shows the recognition results achieved by state-of-the-art methods. As we can observe, our method outperformed existing methods on the AUTSL dataset. This is because we additionally solved visual SR (lip-reading) problems and added features for representing lips. It made it possible to recognize the articulation of speakers in addition to gestures.

## 6. Conclusions

In this article, we present state-of-the-art results on audio-visual speech and gesture recognition. We propose a deep NN-based model architecture for each task. We benchmark our methodology on two well-known datasets: LRW for audio-visual speech recognition and AUTSL for gesture recognition. Results on the LRW dataset show that the proposed model achieves the new state-of-the-art performance on this dataset—98.76% for audio-visual word recognition. Results on the AUTSL dataset demonstrate that the proposed gesture recognition model outperforms existing state-of-the-art and achieves 98.56% gesture recognition performance.

The accuracy of AVSR is achieved by fine-tuning the parameters of both visual and acoustic features and the proposed E2E model. The accuracy of gesture recognition is achieved through the use of a unique set of spatio-temporal features, including those that take into account lip articulation information. Our research integrates two complex tasks in computer vision and machine learning: lip-reading and gesture recognition. A thorough review of prior work reveals that this is the first time lip articulation has been used in the problem of gesture recognition.

The proposed methodology, which is based on NNs, has limitations that are inherent to contemporary machine learning techniques. The limitations are the following:-Data dependency: the performance of both AVSR and SLR methods heavily relies on the quantity and quality of the training data. If the real-world data significantly deviate from the training data, the recognition accuracy will drop significantly.-Sensitivity to noise: in practical applications, both AVSR and SLR methods may encounter acoustic and visual noise that can negatively impact their performance. However, the presence of two information streams (video and audio) provides some level of robustness against noise.-Training time: the proposed NN models require substantial computational resources, making the training process time-consuming. This process involves multiple iterations and calculations in order to optimize the model’s parameters and achieve the desired accuracy. The longer training time not only requires more computational power but also increases the demand for storage and memory resources. Therefore, a trade-off between computational resources, training time, and accuracy should be carefully considered when implementing these models.-Requirement for real-time processing: in order for the proposed AVSR and SLR methods to function in real-time, it is crucial to have access to modern mobile devices equipped with high-performance processors. These powerful devices are necessary to ensure that the NN models can process and analyze the video and audio data quickly and efficiently.

In addition, the evaluation of the speed of AVSR and SLR on portable or mobile devices is crucial for determining the practicality and viability of the proposed methods. The speed is influenced by a multitude of factors, such as the device’s hardware specifications, including the central processing unit (CPU), graphics processing unit (GPU), random access memory (RAM), the neural network architecture, and the pre-processing of data. Our proposed NN models offer real-time capabilities. However, they also require high computational power and a large amount of memory, which can affect their speed of operation on mobile devices. Our evaluation results demonstrate that the proposed AVSR NN model can process a 1.2-s video recording in 0.7 s on mobile devices equipped with an Intel i7 processor. Similarly, our gesture recognition model can process a 2-s video recording in 1.8 s, demonstrating their real-time performance on portable devices. To further optimize the performance of our models and reach a real-time level directly on modern mobile devices, such as the Samsung Galaxy S22, we employed model compression technology with ONNX Runtime. This optimization technique helps reduce the computational and memory demands of the models, allowing them to run smoothly and efficiently on mobile devices.

Furthermore, we conducted a comprehensive evaluation study on how (1) visual model architecture (2DCNN+BiLSTM, 3DCNN, or 3DCNN+BiLSTM), (2) audio model architecture (ResNet-based, VGG-based, or PANN-based), and (3) modalities fusion type (prediction-level, feature-level, or model-level) affect audio-visual speech recognition. We also carefully analyzed the impact of different augmentation techniques on the recognition accuracy and the impact of different dimensionality reduction techniques for gesture recognition and performed model fine-tuning.

We emphasize that the use of visual information can significantly improve speech and gesture recognition. To the best of our knowledge, currently there are no such systems that are able to perform both tasks. The results obtained demonstrate not only the high performance of the proposed methodology, but also the fundamental possibility of recognizing audio-visual speech and gestures by sensors of mobile devices.

Future work in AVSR and SLR recognition will be aimed at enhancing the performance of current algorithms and models. Areas for potential improvement include:-Improving the accuracy and robustness of AVSR and SLR in real-world scenarios where data can be noisy and diverse, and addressing variations in speech and gesture styles, accents, and other sources of variability;-Investigating and creating new models that can effectively handle multilingual and cross-lingual recognition, and demonstrating robust performance across different cultures and dialects.

Overall, there is ample opportunity for growth in the field of AVSR and SLR, and there is a significant demand for innovative approaches, techniques, and technologies to advance the state-of-the-art and make these systems more accessible, user-friendly, and beneficial for a worldwide audience.

## Figures and Tables

**Figure 1 sensors-23-02284-f001:**
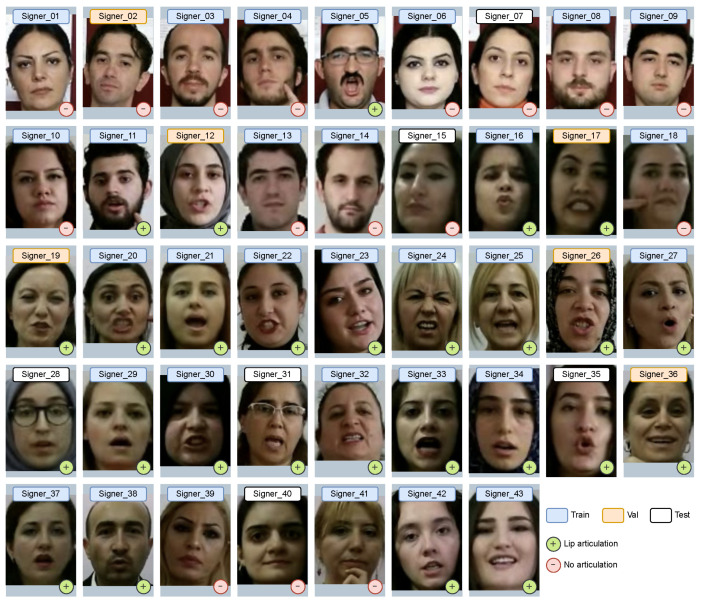
Examples of signers’ faces from the AUTSL dataset.

**Figure 2 sensors-23-02284-f002:**
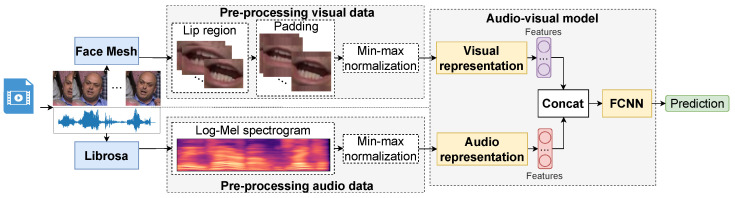
Proposed method for AVSR.

**Figure 3 sensors-23-02284-f003:**
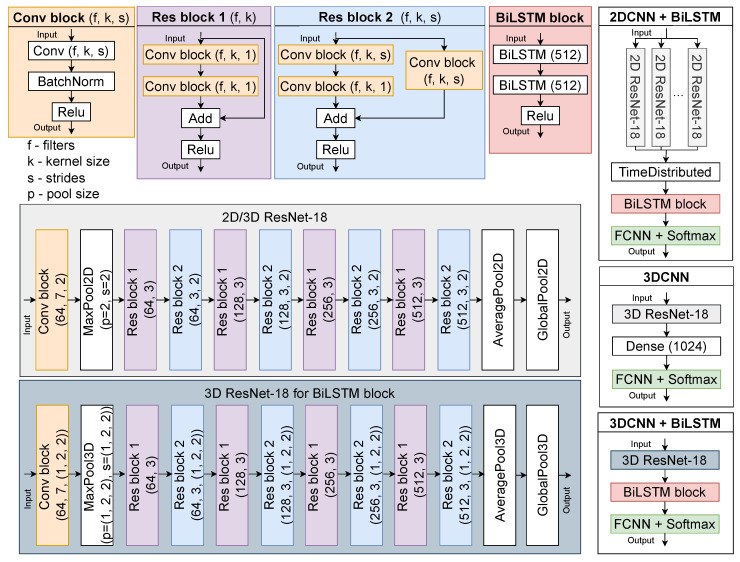
Model architectures for visual speech recognition.

**Figure 4 sensors-23-02284-f004:**
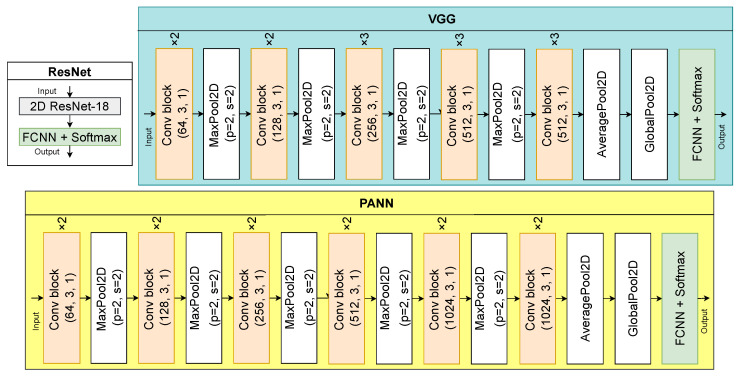
Model architectures for audio speech recognition.

**Figure 5 sensors-23-02284-f005:**
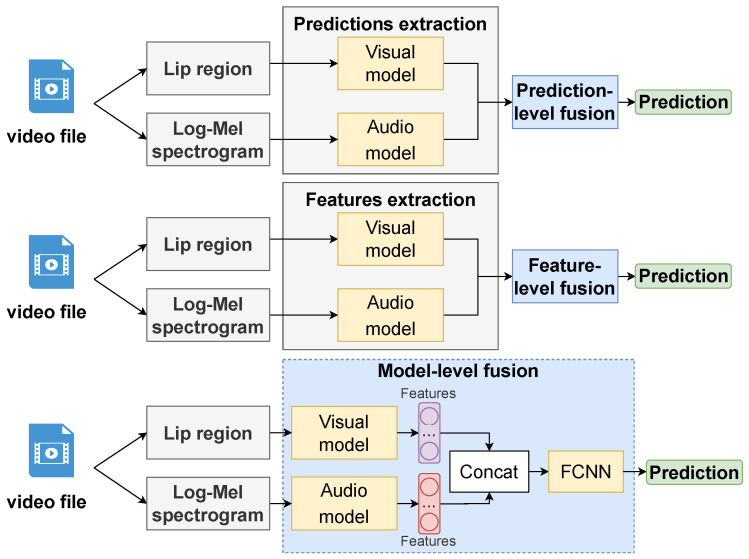
Modalities fusion strategies.

**Figure 6 sensors-23-02284-f006:**
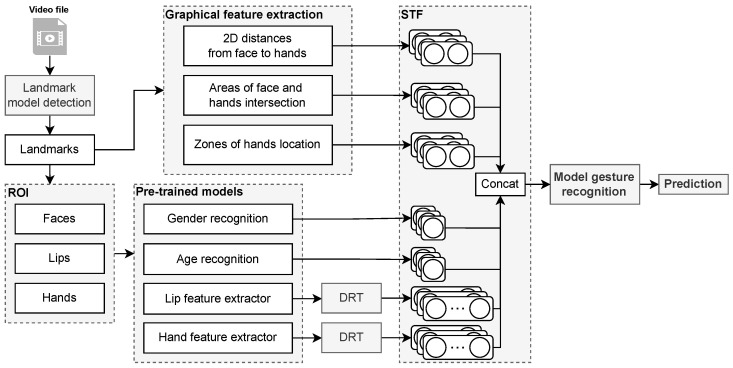
Proposed gesture recognition method. ROI—region-of-interest. DRT—dimensionality reduction technique. STF—spatio-temporal features.

**Figure 7 sensors-23-02284-f007:**
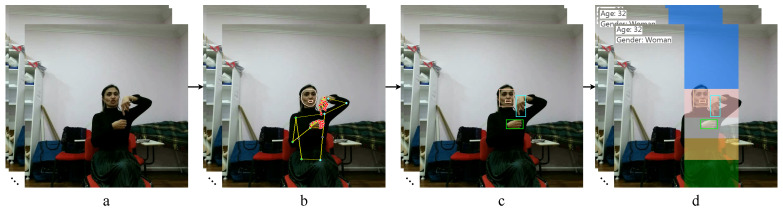
Pipeline for determining graphic regions of interest and gesture zones: (**a**) source frames of the video stream; (**b**) detected landmarks of the face (including lips), hands, and body; (**c**) graphic regions of the face, lips, and hands; (**d**) gesture zones.

**Figure 8 sensors-23-02284-f008:**
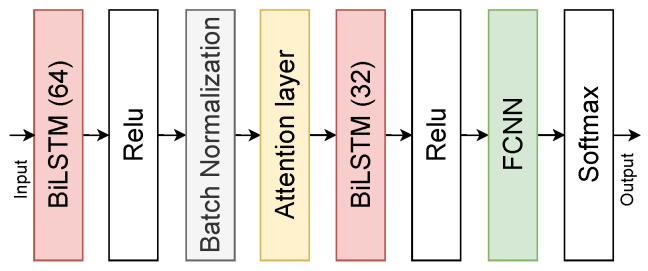
Hand gesture recognition model.

**Table 1 sensors-23-02284-t001:** LRW dataset characteristics.

Set	# Classes	# Samples for Each Class	# Frames
Train	500 (words)	800–1000	29
Val		50	
Test		50	

# Here and in other Tables it means the amount.

**Table 2 sensors-23-02284-t002:** AUTSL dataset characteristics.

Characteristic	Train	Val	Test
Number of signers	31	6	6
Number of articulate signers	19	5	3
Number of gesture repetitions by one signer	1–12	2–6	1–3
Average number of gesture repetitions by one signer	4.0	3.3	2.8
Average gesture repetitions	124.5	19.5	16.6
Number of videos	28,142	4418	3742

**Table 3 sensors-23-02284-t003:** Accuracy results of choosing the optimal visual model.

Model	Optimizer	Learning Rate	Accuracy, %
Constant learning rate			
2DCNN+BiLSTM	Adam	0.0001	83.38
	SGD	0.001	83.10
3DCNN	Adam	0.0001	81.41
	SGD	0.001	81.01
3DCNN+BiLSTM	Adam	0.0001	83.19
	SGD	0.001	82.99
Cosine annealing learning rate			
2DCNN+BiLSTM	Adam	0.0001	**85.35 ***
	SGD	0.001	84.63
3DCNN	Adam	0.0001	83.72
	SGD	0.001	83.51
3DCNN+BiLSTM	Adam	0.0001	85.12
	SGD	0.001	84.39

* Here and in other Tables the best result is highlighted in bold.

**Table 4 sensors-23-02284-t004:** Accuracy results of choosing the optimal input image resolution.

Image Size	# Channels	Image Normalization	Accuracy, %
88 × 88	3	Padding	85.35
88 × 88	1		84.95
112 × 112	3		85.75
44 × 44	3		**86.24**
22 × 22	3		81.00
44 × 44	3	Resize	84.84

**Table 5 sensors-23-02284-t005:** Accuracy results of applying video data augmentation techniques: *p* is the probability (in %) of the maximum number of images to be augmented; α—the smoothing coefficient of the vector.

MixUp, *p*	Label Smoothing, α	Affine Transform, *p*	Accuracy, %
–	–	–	86.24
20	–	–	86.76
40	–	–	80.47
–	0.1	–	86.72
–	0.2	–	86.07
–	–	20	87.03
–	–	40	85.72
20	0.1	20	**87.19**

**Table 6 sensors-23-02284-t006:** Accuracy results of choosing the optimal audio model.

Model	Optimizer	Learning Rate	Accuracy, %
Constant learning rate			
ResNet	Adam	0.0001	91.19
	SGD	0.001	91.86
PANN	Adam	0.00001	70.88
	SGD	0.0001	70.44
VGG	Adam	0.0001	91.15
	SGD	0.0001	91.44
Cosine annealing learning rate			
ResNet	Adam	0.0001	92.04
	SGD	0.001	**92.24**
PANN	Adam	0.00001	84.84
	SGD	0.0001	78.46
VGG	Adam	0.0001	92.08
	SGD	0.0001	91.86

**Table 7 sensors-23-02284-t007:** Accuracy results of choosing the optimal parameters for log-Mel spectrogram.

# Mels	Step Size	Image Size	# Channels	Accuracy, %
128	512	128 × 39	3	92.24
128	512	128 × 39	1	92.77
256	512	256 × 39		91.77
64	512	64 × 39		93.77
64	256	64 × 77		94.45
64	128	64 × 153		94.58
64	64	64 × 305		**95.36**
64	32	64 × 609		95.35
32	64	32 × 305		94.79

**Table 8 sensors-23-02284-t008:** Accuracy results of applying audio data augmentation techniques.

Mixup, *p*	Label Smoothing, α	SpecAugment, *p*	Accuracy, %
–	–	–	95.36
20	–	–	95.59
40	–	–	95.04
–	0.1	–	95.86
–	0.2	–	95.68
–	–	time mask (20)	95.84
–	–	freq mask (20)	95.35
20	0.1	time mask (20)	**96.07**

**Table 9 sensors-23-02284-t009:** The results obtained by the proposed methods of modality fusion in comparison with state-of-the-art results.

SysID	Method	Fusion	Accuracy, %
1	2DCNN + BiLSTM	–	87.16
2	ResNet	–	96.07
3	SysID 1 & 2	Prediction-level	96.87
4	SysID 1 & 2	Feature-level	98.44
5	SysID 1 & 2	Model-level	**98.76**
–	E2E AVSR [50]	Model-level	98.00
–	PBL AVSR [1]	Model-level	98.30

**Table 10 sensors-23-02284-t010:** The results of gesture recognition rate on the test set of the AUTSL dataset when optimizing the dimensionality reduction components for hand configurations.

DRT	# Components			
	2	5	10	15
PCA	87.52	89.60	94.95	95.54
LDA	88.91	92.65	**97.19**	96.82
t-SNE	90.78	–	–	–

**Table 11 sensors-23-02284-t011:** The results of gesture recognition rate on the test set of the AUTSL dataset when optimizing the dimensionality reduction components for lip regions.

DRT	# Components			
	2	5	10	15
For the entire test set				
PCA	98.16	98.40	98.45	98.48
LDA	98.21	**98.56**	98.48	98.32
t-SNE	98.37	–	–	–
For articulating speakers				
PCA	98.98	99.28	99.44	99.54
LDA	99.52	**99.59**	99.48	99.23
t-SNE	99.34	–	–	–

**Table 12 sensors-23-02284-t012:** Comparison of our method with other work (only RGB) on a test set of the AUTSL dataset.

Method	Test Set Recognition Rate, %
Baseline [21]	49.22
De Coster et al. [132]	92.92
Jalba team [146]	96.15
Wenbinwuee team [146]	96.55
Rhythmblue team [146]	97.62
Jiang et al. [133,146]	98.42
**Our**	**98.56**

## Data Availability

In this work we used two large-scale publicly available datasets: LRW– https://www.robots.ox.ac.uk/~vgg/data/lip_reading/lrw1.html, accessed on 30 November 2022, and AUTSL– http://cvml.ankara.edu.tr/datasets, accessed on 30 November 2022. One will need to sign a data sharing agreement with BBC Research & Development before getting access to the LRW dataset.

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
