# Peer review of "Audio-Visual Speech and Gesture Recognition by Sensors of Mobile Devices"

_sensors, 2023, doi:10.3390/s23042284_

Round 1
Reviewer 1 Report
The paper investigates the use of speech, lip reading and gesture to recognize speech in a noisy environment. Standard datasets with augmentation have been used.
The framework for combining the three modalities seems convincing. Elaborate explanation of the experimentation has been provided.
However, it is not mentioned how such a heavy on compute task can be achieved using a mobile device. It would be good to analyse the time and algorithemic complexity of the model to validate the same.
Author Response
We sincerely thank the reviewer for your valuable time and efforts in reviewing our manuscript. Those comments are all valuable and very helpful for revising and improving our paper, as well as the important guiding significance to our researches. We have studied comments carefully and have made corrections which we hope meet with approval.
The description of different fonts used in this document are as follows:
- Reviewers’ original comments are reproduced in red-colored
- Plain fonts are our answers to Reviewers’ comments.
- Text reproduced from the paper is shown in blue color.
Comment 1: However, it is not mentioned how such a heavy on compute task can be achieved using a mobile device. It would be good to analyse the time and algorithemic complexity of the model to validate the same.
Reply: We have expanded the description of how the proposed AVSR and SLR models can be implemented on mobile devices and analysed the time and complexity of the models. We evaluated the execution speed, which turned out to be close to real time on modern mobile devices (e.g., we tested it on Samsung Galaxy S22). The main conclusions regarding the topic are given below. Additional material may also be found in the paper (blue color).
Line 749: In addition, the evaluation of the speed of AVSR and SLR on portable or mobile devices is crucial in determining the practicality and viability of the proposed methods. The speed is influenced by a multitude of factors, such as the device’s hardware specifications, including the Central Processing Unit (CPU), Graphics Processing Unit (GPU), Random Access Memory (RAM), the neural network architecture, and the pre-processing of data. Our proposed NN models offer real-time capabilities. However, they also require high computational power and a large amount of memory, which can affect their speed of operation on mobile devices. Our evaluation results demonstrate that the proposed AVSR NN model can process a 1.2-second video recording in 0.7 seconds on mobile devices equipped with Intel i7 processor. Similarly, our gesture recognition model can process a 2-second video recording in 1.8 seconds, demonstrating their real-time performance on portable devices. To further optimize the performance of our models and reach a real-time level directly on modern mobile devices, such as the Samsung Galaxy S22, we employed model compression technology with ONNX Runtime. This optimization technique helps reduce the computational and memory demands of the models, allowing them to run smoothly and efficiently on mobile devices.

Reviewer 2 Report
Dear Authors,
MAJOR
1. Are there any works available, which analyze the public data sets you used?
MINOR
1. Decipher all abbreviations as they firstly appear (keep the list at the end).
2. Please extend the Conclusions, so that you show more the conclusions, not what you did to get them.
Author Response
We sincerely thank the reviewer for your valuable time and efforts in reviewing our manuscript. Those comments are all valuable and very helpful for revising and improving our paper, as well as the important guiding significance to our researches. We have studied comments carefully and have made correction which we hope meet with approval.
The description of different fonts used in this document are as follows:
- Reviewers’ original comments are reproduced in red-colored
- Plain fonts are our answers to Reviewers’ comments.
- Text reproduced from the paper is shown in blue color.
Comment 1: Are there any works available, which analyze the public data sets you used?
Reply: Indeed, there are many scientific papers analyzing the public databases we use. Regarding audio-visual speech recognition, we used well-known LRW dataset. Regarding gesture recognition, we used AUTSL dataset. We compare the results of our proposed methods with the existing state-of-the-art (Table 9 in the paper for audio-visual speech recognition and Table 12 in the paper for gesture recognition). Proposed methods outperformed existing methods on both tasks. We have achieved AVSR accuracy for the LRW dataset equal to 98.76% and gesture recognition rate for the AUTSL dataset equal to 98.56%.
Comment 2: Decipher all abbreviations as they firstly appear (keep the list at the end).
Reply: We carefully checked all the abbreviations and adjusted them according to your comments. We have also introduced several new abbreviations. We hope that in their current form they have become much more convenient for readers.
Comment 3: Please extend the Conclusions, so that you show more the conclusions, not what you did to get them.
Reply: We have significantly revised the conclusion, focusing on the main findings. The main changes are given below.
Line 721: The accuracy of AVSR is achieved by fine-tuning the parameters of both visual and acoustic features and the proposed E2E model. The accuracy of gesture recognition is achieved through the use of a unique set of spatio-temporal features, including those that take into account lip articulation information. Our research integrates two complex tasks in computer vision and machine learning: lip-reading and gesture recognition. A thorough review of prior work reveals that this is the first time lip articulation has been used in the problem of gesture recognition.
In addition, the evaluation of the speed of AVSR and SLR on portable or mobile devices is crucial in determining the practicality and viability of the proposed methods. The speed is influenced by a multitude of factors, such as the device’s hardware specifications, including the Central Processing Unit (CPU), Graphics Processing Unit (GPU), Random Access Memory (RAM), the neural network architecture, and the pre-processing of data. Our proposed NN models offer real-time capabilities. However, they also require high computational power and a large amount of memory, which can affect their speed of operation on mobile devices. Our evaluation results demonstrate that the proposed AVSR NN model can process a 1.2-second video recording in 0.7 seconds on mobile devices equipped with Intel i7 processor. Similarly, our gesture recognition model can process a 2-second video recording in 1.8 seconds, demonstrating their real-time performance on portable devices. To further optimize the performance of our models and reach a real-time level directly on modern mobile devices, such as the Samsung Galaxy S22, we employed model compression technology with ONNX Runtime. This optimization technique helps reduce the computational and memory demands of the models, allowing them to run smoothly and efficiently on mobile devices.
Furthermore, we conducted a comprehensive evaluation study on how (1) visual model architecture: 2DCNN+BiLSTM, 3DCNN or 3DCNN+BiLSTM, (2) audio model architecture: ResNet-based, VGG-based or PANN-based, (3) modalities fusion type: prediction-level, feature-level or model-level affect audio-visual speech recognition. We also carefully analyzed the impact of different augmentation techniques on the recognition accuracy, the impact of different dimensionality reduction techniques for gesture recognition and performed model fine-tuning.
We emphasize that the use of visual information can significantly improve speech and gesture recognition. To the best of our knowledge, currently there are no such systems that are able to perform both tasks. The results obtained demonstrate not only the high performance of the proposed methodology, but also the fundamental possibility of recognizing audio-visual speech and gestures by sensors of mobile devices.
Future work in AVSR and SLR recognition will be aimed at enhancing the performance of current algorithms and models. Areas for potential improvement include:
- Improving the accuracy and robustness of AVSR and SLR in real-world scenarios where data can be noisy and diverse, and addressing variations in speech and gesture styles, accents, and other sources of variability.
- Investigating and creating new models that can effectively handle multilingual and cross-lingual recognition, and demonstrate robust performance across different cultures and dialects.
Overall, there is ample opportunity for growth in the field of AVSR and SLR, and there is a significant demand for innovative approaches, techniques, and technologies to advance the state-of-the-art and make these systems more accessible, user-friendly, and beneficial for a worldwide audience.

Reviewer 3 Report
In the paper, the authors proposed two deep neural network-based model architectures for audio-visual speech recognition and gesture recognition. The proposed approach has been tested to demonstrate the performance on two well-known datasets: LRW for audio-visual speech recognition and AUTSL for gesture recognition. The results are encouraging.
Strengths:
1. The topic is well defined, not original but the way proposed to tackle this problem is original.
2. The abstract is clear and presents correctly the subject addressed in this paper.
3. The paper contains the basic sections of a scientific paper. It is clear and logically written.
4. The study has been correctly designed.
5. The proposed approach can be applied.
6. The mathematical tool of this paper is correct for me. The hypotheses are correctly identified as such.
7. The English language quality and style of this paper are appropriate and understandable.
Weaknesses:
1. In the abstract part, the novelty of the proposed approach should be described. The authors only described that “This study introduces two deep neural network-based model architectures: one for AVSR and one for gesture recognition. We also incorporate articulation lip-reading for gesture recognition. Since there are no available datasets for the combined task, we evaluated our methods on two different large-scale corpora: LRW and AUTSL, and outperformed existing methods on both tasks audio-visual speech recognition and gesture recognition.” The novelty is not clear. Of course, the reviewer can understand the aim and features of the proposed approach.
2. In the first part of the paper, the authors present various approaches from the literature. Maybe, a synthesis of the solutions proposed depending on the type of analysis, which highlights more clearly the advantages and disadvantages, is useful for readers. This synthesis can be given as a table. Also, the authors should avoid the use of reference loops and present what contains each reference.
3. In the Introduction part, the strong points of the proposed approach should be highlighted in a special paragraph.
4. The limits of the proposed approach should be highlighted better.
5. Future work should be presented.
Author Response
We sincerely thank the reviewer for your valuable time and efforts in reviewing our manuscript. Those comments are all valuable and very helpful for revising and improving our paper, as well as the important guiding significance to our researches. We have studied comments carefully and have made correction which we hope meet with approval.
The description of different fonts used in this document are as follows:
- Reviewers’ original comments are reproduced in red-colored
- Plain fonts are our answers to Reviewers’ comments.
- Text reproduced from the paper is shown in blue color.
Comment 1: In the abstract part, the novelty of the proposed approach should be described. The authors only described that “This study introduces two deep neural network-based model architectures: one for AVSR and one for gesture recognition. We also incorporate articulation lip-reading for gesture recognition. Since there are no available datasets for the combined task, we evaluated our methods on two different large-scale corpora: LRW and AUTSL, and outperformed existing methods on both tasks audio-visual speech recognition and gesture recognition.” The novelty is not clear. Of course, the reviewer can understand the aim and features of the proposed approach.
Reply: We have extended a description of the novelty of the proposed approaches in the abstract. We hope that in the current version the novelty of the proposed methodology is clearer.
Line 8: The main novelty regarding audio-visual speech recognition lies in fine-tuning strategies for both visual and acoustic features and in the proposed end-to-end model, which consider three modality fusion approaches: prediction-level, feature-level and model-level. The main novelty in gesture recognition lies in a unique set of spatio-temporal features, including those that consider lip articulation information.
Comment 2: In the first part of the paper, the authors present various approaches from the literature. Maybe, a synthesis of the solutions proposed depending on the type of analysis, which highlights more clearly the advantages and disadvantages, is useful for readers. This synthesis can be given as a table. Also, the authors should avoid the use of reference loops and present what contains each reference.
Reply: We added the description of each particular reference. We have refrained from providing the literature analysis in the form of a table, since the presented article is based only on recent advances in the field and not the review article. We are afraid that additional descriptions of existing methods would significantly increase the already large volume of the article.
Line 193: The originality of the selected scientific studies is highlighted below.
In [68], presented a method for hand pose estimation and hand shape classification using a multi-layered randomized decision forest algorithm. In a follow-up study [69], the authors proposed a real-time method for capturing hand posture using depth sensors and a 3D hand model with 21 parts and a random decision forest for pixel classification and joint location estimation. Another relevant work [70] proposed a generative model and depth data-based method for hand tracking, using an articulated signed distance function to model hand geometry for fast optimization and high frame rates. This system was capable of tracking two hands interacting with each other or objects.
The work presented in [71] focuses on the development of a real-time CV system for aiding hearing-impaired patients in a hospital setting. The system engages users through a series of questions to determine the purpose of their visit and elicits responses through SL. In [72], the authors propose the use of temporal accumulative features for recognizing isolated SL gestures. This method incorporates SL-specific elements to capture the linguistic characteristics of SL videos, resulting in an efficient and quick SL recognition system. In [73], the authors introduce a method for translating SL into written text using NNs and a learning-based method to tokenization. The authors aim to improve SLR and translation systems by incorporating a tokenization step to better capture the linguistic structure of SL.
In [74], the authors present a method for translating SL into written text using NNs. The study aimed to capture the linguistic structure of SL through a NN-based method, with the findings offering insights into the potential of NNs to improve SLR and translation systems. Another study [75] explores weakly supervised learning for SLR, using a multi-stream CNN-LSTM-HMM model to uncover the sequential parallelism in SL videos. The authors trained the model with weakly labeled data, demonstrating the potential of weakly supervised learning to enhance SLR and translation systems. In [76] address the challenge of multi-articulatory SL translation by proposing a multi-channel Transformer architecture. This architecture enables the modeling of inter- and intra-contextual relationships between different signers, while preserving channel-specific information. The authors of [77], the authors propose an E2E joint architecture based on the Transformer network for SLR and translation. The architecture merges the recognition and translation tasks into a single model, significantly improving performance compared to conventional methods that undertake recognition and translation as separate processes. In [78], the authors examine the challenges of gathering SL datasets for training machine learning models, including privacy, participation, and model performance. The study provides valuable insights into the complexities of collecting high-quality SL data and highlights the importance of considering privacy and ethical concerns in SL research. An interdisciplinary study in [79] provides a comprehensive overview of SL datasets. The study categorizes datasets based on factors like modality, language, and application, and provides an analysis of each dataset and its suitability for various SLR tasks. The authors also discuss the limitations of current datasets and suggest future directions for improvement, making it an important resource for researchers and practitioners in the field of SLR. Also, the authors of [80] presented Microsoft’s submission to the workshop on statistical machine translation shared task on SL translation, which utilized a clean text and full-body Transformer model. The aim of the research was to improve the translation of SL into written text through this methodology.
In [81], the authors provide a comprehensive review of hand gesture recognition techniques, including CV-based methods, machine learning algorithms, and wearable device-based methods. In [82], a method that combines 3DCNN and convolutional LSTM for multimodal gesture recognition is presented, showing the effectiveness of such a combination. The authors in [83] propose a method that improves dynamic hand gesture recognition using 3DCNNs by embedding knowledge from multiple modalities into individual networks. In [84], MultiD-CNN, a multi-dimensional feature learning method for RGB-D gesture recognition using deep CNNs, is proposed. The authors in [85] present a method for gesture recognition using multi-rate and multi-modal temporal enhanced networks, which employs a search algorithm to determine the optimal combination of network architecture, temporal resolution, and modality information. The study in [86] review gesture recognition in robotic surgery, while [87] presents a real-time hand gesture recognition system using YOLOv3, and [88] presents a multi-sensor hand gesture recognition system for teleoperated surgical robots. In [89], the authors show the feasibility of hand gesture recognition using electromagnetic waves and machine learning. The authors in [90] explore ensemble methods for isolated SLR, and in [91], a sign pose-based Transformer method for word-level SLR is proposed. In [92], a SLR method that utilizes a palm definition model and multiple classification is presented, while in [93], an ensemble method using multiple deep CNNs is presented for SLR. In [94], a method for few-shot SLR using online dictionaries is proposed.
Comment 3: In the Introduction part, the strong points of the proposed approach should be highlighted in a special paragraph.
Reply: We have highlighted the strengths of the proposed approaches in the introduction in a special paragraph.
Line 89: In this article, we present state-of-the-art results on audio-visual speech and gesture recognition. We propose a deep neural network-based model architecture for each task. We benchmark our methodology on two well-known datasets: LRW [20] for audio-visual speech recognition and AUTSL [21] for gesture recognition. We outperformed existing methods on both tasks. The accuracy of AVSR is achieved by fine-tuning the parameters of both visual and acoustic features and the proposed E2E model. The accuracy of gesture recognition is achieved through the use of a unique set of spatio-temporal features, including those that take into account lip articulation information. Our research integrates two complex tasks in computer vision and machine learning: lip-reading and gesture recognition. A thorough review of prior work reveals that this is the first-time lip articulation has been used in the problem of gesture recognition.
We emphasize that the use of visual information can significantly improve speech and gesture recognition. To the best of our knowledge, currently there are no such systems that are able to perform both tasks. The results obtained demonstrate not only the high performance of the proposed methodology, but also the fundamental possibility of recognizing audio-visual speech and gestures by sensors of mobile devices.
Comment 4: The limits of the proposed approach should be highlighted better.
Reply: We carefully described the limits of the proposed approaches for both audio-visual speech and gesture recognition. The main conclusions are given below.
Line 728: The proposed methodology, which is based on NNs, has limitations that are inherent to contemporary machine learning techniques. The limitations are following:
- Data dependency: the performance of both AVSR and SLR methods heavily relies on the quantity and quality of the training data. If the real-world data significantly deviates from the training data, the recognition accuracy will drop significantly.
- Sensitivity to noise: in practical applications, both AVSR and SLR methods may encounter acoustic and visual noise that can negatively impact their performance. However, the presence of two information streams (video and audio) provides some level of robustness against noise.
- Training time: the proposed NN models require substantial computational resources, making the training process time-consuming. This process involves multiple iterations and calculations in order to optimize the model’s parameters and achieve the desired accuracy. The longer training time not only requires more computational power but also increases the demand for storage and memory resources. Therefore, a trade-off between computational resources, training time, and accuracy should be carefully considered when implementing these models.
- Requirement for real-time processing: in order for the proposed AVSR and SLR methods to function in real-time, it is crucial to have access to modern mobile devices equipped with high-performance processors. These powerful devices are necessary to ensure that the NN models can process and analyze the video and audio data quickly and efficiently.
Comment 5: Future work should be presented.
Reply: We have added a description of furure work to the article. It will be mainly aimed at enhancing the performance of proposed algorithms and models.
Line 777: Future work in AVSR and SLR recognition will be aimed at enhancing the performance of current algorithms and models. Areas for potential improvement include:
- Improving the accuracy and robustness of AVSR and SLR in real-world scenarios where data can be noisy and diverse, and addressing variations in speech and gesture styles, accents, and other sources of variability.
- Investigating and creating new models that can effectively handle multilingual and cross-lingual recognition, and demonstrate robust performance across different cultures and dialects.
Overall, there is ample opportunity for growth in the field of AVSR and SLR, and there is a significant demand for innovative approaches, techniques, and technologies to advance the state-of-the-art and make these systems more accessible, user-friendly, and beneficial for a worldwide audience.

Round 2
Reviewer 2 Report
All right.